# On Rate-Optimal Partitioning Classification from Observable and from Privatised Data

**Balázs Csanád Csáji**                                                    *csaji@sztaki.hu*
*HUN-REN Institute for Computer Science and Control (SZTAKI);*
*Department of Probability Theory and Statistics,*
*Institute of Mathematics, Eötvös Loránd University (ELTE)*

**László Györfi**                                                          *gyorfi@cs.bme.hu*
*Department of Computer Science and Information Theory,*
*Budapest University of Technology and Economics (BME)*

**Ambrus Tamás**                                                      *ambrus.tamas@sztaki.hu*
*HUN-REN Institute for Computer Science and Control (SZTAKI)*

**Harro Walk**                                    *harro.walk@mathematik.uni-stuttgart.de*
*Institute for Stochastics and Applications, University of Stuttgart*

**Reviewed on OpenReview:** *https: // openreview. net/ forum? id=KYYvIrtgK0*

## Abstract

In this paper we revisit the classical method of partitioning classification and prove novel convergence rates under relaxed conditions, both for observable (non-privatised) and for privatised data. We consider the problem of classification in a $d$ dimensional Euclidean space. Previous results on the partitioning classifier worked with the strong density assumption (SDA), which is restrictive, as we demonstrate through simple examples. Here, we study the problem under much milder assumptions. We presuppose that the distribution of the inputs is a mixture of an absolutely continuous and a discrete distribution, such that the absolutely continuous component is concentrated on a $d_a$ dimensional subspace. In addition to the standard Lipschitz and margin conditions, a novel characteristic of the absolutely continuous component is introduced, by which the convergence rate of the classification error probability is computed, both for the binary and for the multi-class cases. This bound can reach the minimax optimal convergence rate achievable using SDA, but under much milder distributional assumptions. Interestingly, this convergence rate depends only on the intrinsic dimension of the continuous inputs, $d_a$, and not on $d$. Under privacy constraints, the data cannot be directly observed, and the constructed classifiers are functions of the randomised outcome of a suitable local differential privacy mechanism. In this paper we add Laplace distributed noises to the discretisations of all possible locations of the feature vector and to its label. Again, tight upper bounds on the convergence rate of the classification error probability can be derived, without using SDA, such that this rate depends on $2d_a$.

## 1 Introduction

Classification is one of the fundamental problems of machine learning (ML) and mathematical statistics (Devroye et al., 1996; Vapnik, 1998). It has countless applications from health care, agriculture and industry to security, commerce and finance. A significant portion of these applications include sensitive data, for example, about the health or financial circumstances of the clients involved, which should be anonymised before it can be processed. Nevertheless, anonymisation is more involved than just removing the names and the addresses of the clients, as the data can contain several other types of sensitive information. The

concept of differential privacy (Dwork et al., 2006) provides a rigorous framework to measure the amount of information privacy. Local privacy dates back to (Warner, 1965), and *local differential privacy* (LDP) was formally defined by Duchi et al. (2013). On the one hand, LDP helps to get rid of a trusted third party; on the other hand, it generates a framework for managing nonparametric regression and classification, cf. (Berrett & Butucea, 2019) and (Berrett et al., 2021). The LDP mechanism allows processing the data even in cases when the original, raw data should only be seen by its legitimate data holder.

In this paper, we focus on partitioning rules which represent archetypical classification methods, see (Kohler & Krzyżak, 2007). They are key components of several machine learning techniques, such as decision trees, random forests, piecewise estimators and hierarchical models. Their ability to tackle scalability, interpretability, and explainability challenges makes them well-suited for integration into ML frameworks such as federated learning (Argente-Garrido et al., 2025) and ensemble-based methods (Chen & Guestrin, 2016). They are especially useful in low-dimensional settings and provide benchmark models for complex data structures. Additional applications include hyperparameter tuning and data cleaning. Local averaging estimates are also commonly used to prove theoretical results in nonparametric settings, e.g., rate of convergence and minimax theorems. Our aim is to analyse the *convergence rate* of *classification error probability* associated with partitioning rules under mild statistical assumptions. We investigate both binary and multi-class classification, covering observable and privatised (anonymised) data. In the latter case, we apply Laplace type randomisation which ensures LDP constraints, cf. (Berrett & Butucea, 2019).

Partitioning classification is often studied under *Lipschitz* and *margin* types conditions, but the existing optimal error bounds (for the classical, non-privatised case) suppose a further condition, called the *strong density assumption* (SDA). This assumption, stated in Equation 5, ensures that the probability measure of each cell of the partition is bounded away from zero. For observable (non-privatised) data, Kohler & Krzyżak (2007) studied the convergence rate of plug-in partitioning classification with and without SDA. Their rate with SDA was proven to be *minimax optimal* by Audibert & Tsybakov (2007). Furthermore, for the privatised case, Berrett et al. (2021) *conjectured* that in the absence of such an assumption, the convergence rate could be *arbitrarily slow*. One of our main contributions is to show that this conjecture is incorrect by establishing fast convergence rates for privatised partitioning rules without applying the SDA.

We argue that the SDA is fairly restrictive, as demonstrated through Examples 1, 2 and 3, for which the SDA is not satisfied. In general, the Lipschitz and the margin parameters do not determine the convergence rate of the error probability. While the approximation error (bias) depends only on the Lipschitz and margin parameters, the estimation error is very sensitive to the behaviour of the distribution of the feature vector around the decision boundary. These motivate the introduction of an additional parameter characterising the relation between the margin and the low density areas. Specifically, we assume that the distribution of input $X$ is a mixture of an absolutely continuous and a discrete distribution, such that the absolutely continuous component is supported on a $d_a$-dimensional subspace. This assumption is natural in many applications, as real-world data often combine continuous variables with categorical data (e.g., in healthcare, continuous physiological measurements together with discrete diagnostic codes or demographic data; in finance, asset returns combined with categorical credit ratings). For the absolutely continuous component, we introduce an additional mild condition, called the *combined margin and density assumption*. The proposed convergence rate for the estimation error incorporates the combined margin and density parameter, which is an appropriate characterisation of the intersection of the low density region and the decision boundary.

Our convergence rate for the binary case, proved *without* the *SDA*, greatly improves the SDA independent rate of Kohler & Krzyżak (2007), see Equation 6, as it is demonstrated on our specific examples with appropriately chosen parameters. Our result ensures the same optimal rate for Example 1 as the *SDA dependent* rate of Kohler and Krzyżak, cf. Equation 7. This demonstrates that the optimal rate can be achieved even without the SDA. Moreover, for Example 2 our bound matches the convergence rate one gets by direct calculation of the rate, showing that the bound is *tight*. Finally, for Example 3, though the deduced SDA independent convergence rate is worse than the SDA dependent rate of Equation 7, but it still improves the previously known SDA independent rate, that is Equation 6. Furthermore, interestingly, this convergence rate only depends on the intrinsic dimension of the absolutely continuous part, $d_a$, and not on the dimension of the whole $X$. We also derive upper bounds for the convergence rate for the case of privatised data with LDP guarantees, which rate depends on $2d_a$, instead of $d_a$, that was the case for the nonprivate bound. Both of

these bounds are extended to multi-class classification. Our bounds for privatised partitioning classification refute the conjecture that without the SDA the convergence rate could be arbitrarily slow.

The structure of the paper is as follows. First, in Section 2, we revisit and improve the error probability bounds of partitioning classifiers for observable data, both for binary and multi-class setups. Then, in Section 3, we follow an analogous program for privatised partitioning classifiers. The results are summarised and discussed in Section 4. The detailed proofs are presented in the Appendix.

## 2 Partitioning classification from observable data

In this section, we study the problem of classification from observable (non-privatised) data. First, we give an overview of the core problem, recall the partitioning classification rule, and state our main assumptions under which we quantify the convergence rate both for binary and for multi-class classification.

The standard setup of binary classification is as follows: let the random feature vector $X$ take values in $\mathbb{R}^d$, and let its label $Y$ be $\pm 1$ valued. We denote by $\mu$ the distribution of $X$, that is, $\mu(A) = \mathbb{P}(X \in A)$ for all measurable sets $A \subseteq \mathbb{R}^d$. The task of classification is to decide on $Y$ given $X$, i.e., one aims to find a decision function $D$ defined on the range of $X$ such that $D(X) = Y$ with large probability. If $D$ is an arbitrary (measurable) decision function, then its *error probability* is denoted by

$$L(D) = \mathbb{P}\{D(X) \neq Y\}.$$

Let us denote the *regression function* by

$$m(x) = \mathbb{E}[Y \mid X = x],$$

which is well-defined for $\mu$-almost all $x \in \mathbb{R}^d$. It is well-known that the Bayes decision function defined by

$$D^*(x) = \text{sign}(m(x)),$$

where $\text{sign}(x) = \mathbb{I}_{\{x \geq 0\}} - \mathbb{I}_{\{x < 0\}}$ with indicator function $\mathbb{I}$, minimizes the error probability. Then,

$$L^* = \mathbb{P}\{D^*(X) \neq Y\} = \min_D L(D)$$

denotes the minimal error probability (i.e., the probability of misclassification).

Let $\mathcal{P}_h = \{A_{h,1}, A_{h,2}, \ldots\}$ be a cubic partition of $\mathbb{R}^d$ with cubic cells $A_{h,j}$ of volume $h^d$ such that $(0, h]^d \in \mathcal{P}_h$. Data $\mathcal{D}_n$ is assumed to contain independent identically distributed (i.i.d.) copies of the random vector $(X, Y)$,

$$\mathcal{D}_n = \{(X_1, Y_1), \ldots, (X_n, Y_n)\}. \tag{1}$$

Let

$$\nu_n(A_{h,j}) = \frac{1}{n} \sum_{i=1}^n Y_i \, \mathbb{I}_{\{X_i \in A_{h,j}\}}.$$

The well-known *partitioning classification* rule is

$$D_n(x) = \text{sign}(\nu_n(A_{h,j})), \quad \text{if } x \in A_{h,j}. \tag{2}$$

The main goal of this paper is to prove novel convergence rates for the expected excess risk, $\mathbb{E}\{L(D_n)\} - L^*$. Furthermore, we extend these results to the multi-class setting and provide generalizations for both binary and multi-class privatised versions of $D_n$.

A nontrivial rate of convergence of any classification rule can be derived under some smoothness condition. The *Lipschitz condition* on $m$ means that there is a constant $C$ such that for all $x, z \in \mathbb{R}^d$, we have

$$|m(x) - m(z)| \leq C \, \|x - z\|. \tag{3}$$

Concerning the rate of convergence of any classification rule, Mammen & Tsybakov (1999) and Tsybakov (2004) discovered and investigated the phenomenon that there is a dependence on the behaviour of $m$ in the neighbourhood of the decision boundary

$$B^* = \{x : m(x) = 0\}.$$

The *margin condition* means that for all $0 < t \leq 1$, we have

$$G^*(t) := \int \mathbb{I}_{\{0 < |m(x)| \leq t\}} \mu(dx) \leq c^* \cdot t^\gamma, \tag{4}$$

for some constants $c^* \geq 0$ and $\gamma \geq 0$. The margin condition holds trivially for $\gamma = 0$. If $\gamma$ is greater, then the condition is more restrictive. Our convergence rates will be proportianal to $\gamma$, thus, the proven convergence will be faster for larger values of $\gamma$.

The *strong density assumption* (SDA) holds, when for all $\mu(A_{h,j}) > 0$, we have

$$\mu(A_{h,j}) \geq ch^d, \qquad j = 1, \ldots, \tag{5}$$

for some constant $c > 0$. The SDA is a restrictive condition, because, for example, it excludes continuous densities reaching zero, see the examples below. We establish novel bounds for these cases, as well.

If $X$ is bounded, and the margin and the Lipschitz conditions on $m$ are satisfied, then Kohler & Krzyżak (2007) showed for partitioning classification with suitably chosen $(h_n)$ that

$$\mathbb{E}\{L(D_n)\} - L^* = O\left(n^{-\frac{1+\gamma}{3+\gamma+d}}\right). \tag{6}$$

If, in addition, the SDA is met, then the order of the rate of convergence is

$$n^{-\frac{1+\gamma}{2+d}}. \tag{7}$$

Under the SDA, Audibert & Tsybakov (2007) proved the *minimax optimality* of this rate, i.e., they showed that this rate is also a lower bound for any classification rule over the class of distributions satisfying the aforementioned conditions. However, the Lipschitz and margin conditions do not determine the true rate of convergence of the error probability.

Let us consider three examples. It is easy to see that none of these examples satisfy the SDA condition, as the densities are not bounded away from zero.

**Example 1.** *Let the range of $X$ be the interval $[-1, 1]$ and*

$$m(x) = x, \quad |x| \leq 1.$$

*Furthermore, assume that $\mu$ has the density*

$$f(x) = c_\delta(1 - |x|^\delta), \quad |x| < 1,$$

*with $\delta > 0$ and $c_\delta$ being a normalising constant.*

**Example 2.** *Let $m$ be as in Example 1. Assume that $\mu$ has the density*

$$f(x) = c_\delta|x|^\delta, \quad 0 < |x| \leq 1,$$

*with $\delta > 0$ and normalising constant $c_\delta$.*

**Example 3.** *Let the range of $X$ be the interval $[-1, 1]$ and*

$$m(x) = \text{sign}(x) \cdot x^2, \quad |x| \leq 1.$$

*Assume that $\mu$ has the density*

$$f(x) = |x|, \quad |x| \leq 1.$$

In Example 1 the margin condition holds with $\gamma = 1$. In the literature only the suboptimal rate (6) has been proven, which is $n^{-2/5}$. By an easy calculation, for the choice $h_n = n^{-1/3}$, we obtain that the true rate is $n^{-2/3}$, which corresponds to the optimal rate (7). This means that there is space for improvement. In this example the boundary of the regression function is separated from the boundary of the density. In Examples 2 and 3 the margin condition holds with $\gamma = \delta + 1$ and $\gamma = 1$, respectively, however, in both cases the density vanishes at the decision boundary of $m$, making it hard to control the risk of a plug-in classifier. In order to address this problem, we introduce a new condition that combines the density and regression functions. This condition characterizes the relation between the margin and the low density areas with an additional parameter. Interestingly, if $-1 < \delta \leq 0$ for the density in Example 2, then SDA holds and the minimax optimal rate of Audibert & Tsybakov (2007) is achieved by the partitioning rule.

Let us assume that the distribution of the inputs, $\mu$, can be decomposed as

$$\mu = \mu_a + \mu_s, \tag{8}$$

where $\mu_a$ is an *absolutely continuous* distribution on its support $S_a$, which is contained in a (possibly unknown) $d_a$ dimensional Euclidean space. We denote the density of $\mu_a$ with respect to the $d_a$ dimensional Lebesgue measure $\lambda_{d_a}$ by $f$. Furthermore, assume that $\mu_s$ is a *discrete* distribution with support $S_s$ of finite size. For this setup, our motivation was the example when $X$ has $d_a$ absolutely continuous coordinates and $d_s$ discrete features. One of the main goals of this paper is to allow continuous densities with our novel combined condition, which do not necessarily admit the SDA.

Under the Lipschitz and margin conditions, tight bounds can be derived for the approximation error component of the error probability. We introduce a novel condition on the relationship between the margin area and the low-density region. This will be a useful tool for the refined convergence rate analysis. Let

$$f_h(x) = \frac{\mu_a(A_{h,j})}{\lambda_{d_a}(A_{h,j})} = \frac{\int_{A_{h,j}} f(z)\lambda_{d_a}(dz)}{\lambda_{d_a}(A_{h,j})}, \quad \text{if } x \in A_{h,j}. \tag{9}$$

One can observe that function $f_h$ is the expectation of the histogram density estimate. Additionally, if $\mu$ is absolutely continuous w.r.t. the Lebesgue measure $\lambda_d$, then under the SDA for $\mu$-almost all $x$ we have

$$f_h(x) = \frac{\mu(A_{h,j})}{\lambda_d(A_{h,j})} \geq c. \tag{10}$$

Our new *combined margin and density condition*, introduced below, characterizes the measure of those regions which are either close to the decision boundary or for which the density of $X$ is small. Assume that there exists $h_1^* > 0$ such that for any $h \in (0, h_1^*)$ and for all $t > 0$, we have

$$G_h(t) := \int_{S_a} \mathbb{I}_{\{0 < \sqrt{f_h(x)}|m(x)| \leq t\}} \frac{f(x)}{\sqrt{f_h(x)}} \lambda_{d_a}(dx) \leq c_1 \cdot t^{\gamma_1} \tag{11}$$

with constants $c_1 > 0$ and $0 \leq \gamma_1 = \gamma_1(\sqrt{f})$. Similarly to the margin condition (4), the combined margin and density condition (11) becomes more restrictive if $\gamma_1$ increases. One of our main results is that the convergence rate is controlled by the minimum of $\gamma$ and $\gamma_1$. In general the margin condition with $\gamma$ is not stronger than the combined margin and density condition with the same $\gamma$ and vice versa. However, if $X$ is an absolutely continuous random vector in $\mathbb{R}^d$, then it is easy to see that the SDA and the margin condition with $\gamma \geq 0$ implies the combined margin and density condition for $\gamma_1 \leq \gamma$. Hence, in this case our novel combined condition and the margin condition together are weaker than the SDA and the margin condition. The following lemma considers the case when SDA is assumed only in the neighbourhood of the decision boundary, i.e., when we have $f_h(x) \geq f_{\varepsilon,min} > 0$ around the decision boundary. This together with the margin condition with $\gamma \geq 0$ is sufficient to prove that the combined margin and density condition holds for every $\gamma_1 \leq \min(\gamma, 1)$. The proof of Lemma 2.1 is included in Appendix A.1.

**Lemma 2.1.** *For $0 < \epsilon < 1$ and $0 < h_0$, set*

$$B_\epsilon^* = \{x : |m(x)| \leq \epsilon\},$$

*and $f_{\epsilon,min} = \inf_{x \in B_\epsilon^*, 0 < h < h_0} f_h(x)$. If there is an $\epsilon \in (0,1)$ such that $f_{\epsilon,min} > 0$ and the margin condition holds with $\gamma$, then the combined margin and density condition holds for every $\gamma_1 \leq \min(\gamma, 1)$.*

The generalised Lebesgue density theorem yields that for $\lambda_{d_a}$-almost all $x$,

$$\lim_{h \downarrow 0} f_h(x) = f(x), \tag{12}$$

cf. Theorem 7.2 in (Wheeden & Zygmund, 1977). Because of Equation 12, we *conjecture* that if

$$\int_{S_a} \mathbb{I}_{\{0 < \sqrt{f(x)}|m(x)| \leq t\}} \sqrt{f(x)} \lambda_{d_a}(dx) \leq \tilde{c} \cdot t^{\gamma_1}$$

with a $\tilde{c} > 0$ and $\gamma_1 \geq 0$, then the combined margin and density condition holds.

The following theorem establishes new upper bounds on the convergence rate of the error probability without requiring the SDA condition. Interestingly, only dimension $d_a$ matters, hence if the absolutely continuous component is concentrated within a low dimensional subspace, then the convergence is fast. The proof of Theorem 2.2 is presented in Appendix A.2.

**Theorem 2.2.** *Assume that $X$ is bounded, $m$ satisfies the Lipschitz condition, Equation 3, and the margin condition, Equation 4 with $\gamma \geq 0$. In addition, the combined margin and density condition of Equation 11 holds with $\gamma_1 \geq 0$. Let $(h_n)$ be a monotonic decreasing sequence with zero limit. Then, we have*

$$\mathbb{E}\{L(D_n)\} - L^* = O\left(h_n^{1+\gamma}\right) + O\left((nh_n^{d_a})^{-(1+\min\{1,\gamma,\gamma_1\})/2}\right). \tag{13}$$

The main steps of the proof are as follows. First, we decompose the expected excess risk of the plug-in classifier, defined by (2), into an *approximation error* and an *estimation error*. The Lipschitz continuity of $m$, combined with the margin condition, yields a polynomial bound of order $h^{1+\gamma}$ for the approximation error. For the estimation error, we use a central limit theorem (CLT) based approximation of the cell averages, which provides a Gaussian-type exponential tail bound. Then, we rewrite the resulting integrals via the Lebesgue-Stieltjes representation, thereby reducing the spatial integral to a single dimension. Finally, the margin condition and the combined margin and density condition yield a polynomial rate in $nh^d$.

For known $d_a$ and for the choice

$$h_n = n^{-\frac{1}{2+d_a}}, \tag{14}$$

one has that

$$\mathbb{E}\{L(D_n)\} - L^* = O\left(n^{-\frac{1+\min\{1,\gamma,\gamma_1\}}{2+d_a}}\right). \tag{15}$$

If $d_a$ is not known, then choosing $h_n = n^{-\frac{1}{2+d}}$ yields a slightly worse bound, where the first term is $O\left(n^{-\frac{1+\gamma}{2+d}}\right)$.

Note that if $\gamma \leq \min(1, \gamma_1)$, then our bound achieves the *minimax optimal* rate of Audibert & Tsybakov (2007) under milder assumptions than the SDA. The bounded support condition ensures that the partition contains only $O(h^{-d_a})$ cells with positive probability. This is essential for bounding the estimation error; see (30) and (31). Extensions to unbounded supports would require control of the tail behaviour of $\mu_a$.

In Example 1 for all $\delta > 0$, the margin condition and the combined margin and density condition hold such that $\gamma = 1$ and for all $\gamma_1 \leq \gamma$ because of Lemma 2.1. Thus, (15) results in the rate $n^{-2/3}$, which is the same as the optimal rate in (7). For Example 2, one can use $\gamma = \delta + 1$ and $\gamma_1 = 1$, therefore the bound on the rate in (15) is equal to $n^{-2/3}$ for $\delta > 0$. For $-1 < \delta \leq 0$, when the SDA holds, our rate is $n^{-(2+\delta)/3}$, which is the same as the one proved by Audibert & Tsybakov (2007). In Example 3, one has $\gamma = 1$ and $\gamma_1 = 3/5$ and so $\gamma > \gamma_1$. Thus, the rate in Theorem 2.2 is $n^{-8/15}$, which is faster than the poor rate in Equation 6, but it is worse than the rate in Equation 7 with $\gamma = 1$.

The combined margin and density condition always holds with $\gamma_1 = 0$. Then, using (14), by (15) we have

$$\mathbb{E}\{L(D_n)\} - L^* = O\left(n^{-\frac{1}{2+d_a}}\right).$$

In (13) the first term, called the *approximation error* bound, follows from the Lipschitz condition and from the margin condition, while the second term, called the *estimation error* bound, has been derived from the

margin condition and from the combined margin and density condition. Note that the upper bound on the estimation error is managed by the CLT approximation with an error term of order $O(1/(nh_n^{d_a}))$. Therefore, due to the CLT approximation, super-fast rates are not achieved when $\min(\gamma, \gamma_1) \geq 1$.

Next, we consider the multi-class classification problem in which case $Y$ takes values in $\{1, \ldots, M\}$. Let

$$P_k(x) = \mathbb{P}\{Y = k \mid X = x\}$$

denote the a posteriori probabilities for $k = 1, \ldots, M$. Then, the Bayes decision has the form

$$D^*(x) = \arg\max_k P_k(x).$$

Let $P_{n,k}$ be an estimate of $P_k$ based on $\mathcal{D}_n$. Then, the plug-in classification rule $D_n$ derived from $P_{n,k}$ is

$$D_n(x) = \arg\max_k P_{n,k}(x).$$

Recently, Xue & Kpotufe (2018) and Puchkin & Spokoiny (2020) generalised the margin condition to the multi-class setting: let $P_{(1)}(x) \geq \cdots \geq P_{(M)}(x)$ be the ordered values of $P_1(x), \ldots, P_M(x)$. For multiple classes, the *margin condition* means that there are some $\gamma \geq 0$ and $c^* \geq 0$ such that

$$G^*(t) := \int \mathbb{I}_{\{0 < P_{(1)}(x) - P_{(2)}(x) \leq t\}} \mu(dx) \leq c^* t^\gamma \quad \forall\, t > 0. \tag{16}$$

Using this concept of margin condition, Györfi & Weiss (2021) computed the rate of convergence of a nearest neighbour based prototype classifier, when the feature space is a separable metric space.

For multiple classes, the *combined margin and density condition* means that there is a $h_1^* > 0$ such that for any $h \in (0, h_1^*)$ and for all $0 < t$, we have

$$G_h(t) := \int_{S_a} \mathbb{I}_{\{0 < \sqrt{f_h(x)}(P_{(1)}(x) - P_{(2)}(x)) \leq t\}} \frac{1}{\sqrt{f_h(x)}} \mu_a(dx) \leq c_1 \cdot t^{\gamma_1} \tag{17}$$

with constants $0 < c_1$ and $0 \leq \gamma_1$.

The multi-class partitioning rule is defined by

$$\nu_{n,k}(A_{h,j}) = \frac{1}{n} \sum_{i=1}^n \mathbb{I}_{\{Y_i = k, X_i \in A_{h,j}\}},$$

and the corresponding plug-in rule as

$$D_n(x) = \arg\max_k \nu_{n,k}(A_{h,j}) \quad \text{for } x \in A_{h,j}.$$

Our main result for the multi-class problem is as follows. Its proof is presented in Appendix A.3.

**Theorem 2.3.** *Assume that $X$ is bounded. Additionally, assume that $P_1, \ldots, P_M$ satisfy the Lipschitz condition, Equation 3, the margin condition, Equation 16 with $\gamma \geq 0$ and the combined margin and density condition, Equation 17 with $\gamma_1 \geq 0$. Let $(h_n)$ be a monotonically decreasing sequence with zero limit. Then,*

$$\mathbb{E}\{L(\widetilde{D}_n)\} - L^* = O\left(M^2 h_n^{1+\gamma}\right) + O\left(M^2 (nh_n^{d_a})^{-(1+\min\{1,\gamma,\gamma_1\})/2}\right).$$

Similarly as above for $h_n = n^{-1/(2+d_a)}$ we have the rate of (15). This is the first convergence rate result for multi-class plug-in classifiers without the SDA using only margin-type conditions. The key step of proving Theorem 2.3, beside the CLT approximation, is the application of Lemma A.1, which is an extension of (Györfi & Weiss, 2021, Lemma 8). The bound grows quadratically with the number of classes.

## 3 Partitioning classification under local differential privacy

One of the main purposes of this paper is to bound the error probability of partitioning classifiers in the case, when the raw data $\mathcal{D}_n$ is not directly accessible, but only a suitably anonymised surrogate. More precisely, the anonymised data must satisfy a *local differential privacy* (LDP) condition (Berrett & Butucea, 2019; Duchi et al., 2013). Our work is motivated by (Berrett & Butucea, 2019), where the first step in this direction was done. We note that the same privatisation mechanism was studied for regression and density estimation in (Györfi & Kroll, 2025) and (Györfi & Kroll, 2023), respectively.

Let us now state the privacy mechanism that we consider in this work for the anonymisation of the raw data $\mathcal{D}_n$. Our approach follows the technique of Laplace perturbation already considered in (Berrett & Butucea, 2019). In this privacy setup, the data holder of $X_i$ generates and transmits the data

$$Z_{i,j} = Y_i \mathbb{I}_{\{X_i \in A_{h,j}\}} + \sigma_Z \epsilon_{i,j}, \quad j = 1, \ldots \tag{18}$$

to the statistician, where the noise level is $\sigma_Z > 0$, and $\{\epsilon_{i,j}\}$ ($i = 1, \ldots, n$, $j = 1, \ldots$) are independent centred Laplace random variables with unit variance. This means that individual $i$ generates noisy data for every cell $A_{h,j}$. We can observe that this privacy mechanism is locally differential, because each set of $\{Z_{i,j}\}, j = 1, \ldots$, can be computed separately, i.e., no other $Z_{k,j}$ with $k \neq i$ is used in the privatisation. The Laplace mechanism is particularly well-suited in this context, as its exponential ($\ell_1$-based) form directly aligns with the likelihood ratio constraint, yielding exact privacy guarantees that are easy to calibrate.

Now, we briefly recall the definition of LDP. Non-interactive privacy mechanisms can be described by the conditional distributions $Q_i$ of the privatised data $Z_i$, for $i = 1, \ldots, n$, where each $Z_i$ takes its values from a measurable space $(\mathcal{Z}, \mathscr{Z})$. Specifically, given a realisation of the raw data $(X_i, Y_i) = (x_i, y_i)$, one generates $Z_i$ according to the probability measure defined by $Q_i(A \mid (X_i, Y_i) = (x_i, y_i))$, for any $A \in \mathscr{Z}$. Such a non-interactive mechanism is *local* since any data holder can independently generate privatised data (e.g., without a trusted third party). For a privacy parameter $\alpha \in [0, \infty]$, a non-interactive privacy mechanism is said to be an $\alpha$-*locally differentially private* mechanism if the condition

$$\frac{Q_i(A \mid (X_i, Y_i) = (x, y))}{Q_i(A \mid (X_i, Y_i) = (x', y'))} \leq \exp(\alpha)$$

holds for all $A \in \mathscr{Z}$ and all realisations $(x, y)$, $(x', y')$ of the raw data. The noise level $\sigma_Z$ in (18) has to be chosen of the form $2\sqrt{2}/\alpha$, to make the overall mechanism satisfy $\alpha$-LDP (Berrett & Butucea, 2019).

For privatised data, Berrett & Butucea (2019) introduced the privatised partitioning estimator

$$\widetilde{\nu}_n(A_{h,j}) = \frac{1}{n} \sum_{i=1}^{n} Z_{i,j}, \quad \text{if } x \in A_{h,j},$$

and the corresponding plug-in classifier

$$\widetilde{D}_n(x) = \operatorname{sign} \widetilde{\nu}_n(A_{h,j}), \quad \text{if } x \in A_{h,j}.$$

If $h = h_n \to 0$ and $\alpha = \alpha_n \to 0$ such that $n\alpha_n^2 h_n^{2d} \to \infty$, then Berrett & Butucea (2019) proved the universal consistency of the partitioning classifier and calculated the minimax rate

$$(n\alpha_n^2)^{-\frac{1+\gamma}{2(1+d)}}$$

in the class, when the margin condition and the Lipschitz condition together with SDA hold.

For a fixed partition, computing the empirical cell averages for the nonprivate partitioning estimate requires $O(n)$ operations. For bounded inputs, storage and prediction scale with the number of occupied cells, which is $O(h^{-d_a})$. In the private version, additional computation is required to generate and aggregate noise at the cell level. Hence, the effective workload scales with the number of cells, that is, $O(nh^{-d_a})$.

Again, instead of the SDA we rely on novel margin-type condition. Let us introduce the modified combined margin and density condition for privatisation. We say that $m$ satisfies the modified combined margin and density condition if there exists $h_2^*$ such that for all $h \in (0, h_2^*)$ we have for all $t > 0$:

$$\widetilde{G}_h(t) \doteq \int \mathbb{I}_{\{0 < f_h(x)|m(x)| \leq t\}} \frac{1}{f_h(x)} \mu_a(dx) \leq c_2 \, t^{\gamma_2} \tag{19}$$

with $c_2 > 0$ and $\gamma_2 \geq 0$. We note that $f_h$ is used here, instead of $\sqrt{f_h}$, because an extra term of 2 appears for privatisation, similarly as in the bound of Berrett & Butucea (2019). It is easy to prove that the modified combined margin and density condition is less restrictive than the SDA and the margin condition together.

In Example 1 the modified condition holds for every $\gamma_2 \leq 1$. For Example 2 the modified margin and density condition is satisfied with $\gamma_2 = 1/(\delta + 1)$, while for Example 3, (19) holds with $\gamma_2 = 1/(\delta + 2)$. In general it cannot be proved that the modified combined margin and density condition is more restrictive than the combined margin and density condition. However, if the SDA holds, then we can prove the lemma that follows, see Appendix A.4:

**Lemma 3.1.** *For $0 < h_0$ let $f_{min} = \inf_{x \in S_a, 0 < h < h_0} f_h(x)$. If $f_{min} > 0$ and the combined margin and density condition holds with $\gamma_1$, then the modified combined margin and density condition holds for every $\gamma_2 \leq \gamma_1$.*

Interestingly, the margin condition and the modified combined margin and density condition together is more restrictive than the original combined margin and density condition. The proof of Lemma 3.2 can be found in Appendix A.5.

**Lemma 3.2.** *If the margin condition holds with $\gamma \geq 0$ and the modified combined margin and density condition holds with $\gamma_2 \geq 0$, then the combined margin and density condition holds for every $\gamma_1 \leq \min(\gamma_2, \gamma)$.*

The next theorem is the extension of Theorem 2.2 to locally differentially private partitioning classifiers. It is proved in Appendix A.6.

**Theorem 3.3.** *Assume that $X$ is bounded, $m$ satisfies the Lipschitz condition, Equation 3, the margin condition, Equation 4 with $\gamma \geq 0$, the combined margin and density condition, Equation 11 with $\gamma_1 \geq 0$, and the modified combined margin and density condition, Equation 19 with $\gamma_2 \geq 0$. Let $(h_n)$ be a monotonically decreasing sequence with zero limit. Then,*

$$\mathbb{E}\{L(\widetilde{D}_n)\} - L^* = O\left(h_n^{1+\gamma}\right) + O\left(\left(\frac{1}{nh_n^{d_a}}\right)^{(1+\min\{1,\gamma_1\})/2}\right) + O\left(\left(\frac{\sigma_Z^2}{nh_n^{2d_a}}\right)^{(1+\min\{\gamma,\gamma_2\})/2}\right) + O\left(\frac{\sigma_Z}{nh_n^{d_a}}\right).$$

The expected excess risk consists of the usual approximation and estimation error terms from the nonprivate case, plus additional privatisation terms, see Equation 39 in the proof, that reflect the probability that the privatisation noise flips the sign of the decision function. In the proof, for this privatisation term, using the Berry-Esseen theorem, we obtain a Gaussian tail bound, with a remainder of order $O(\sigma_Z/(nh^{d_a}))$. The tail is then controlled, as in the non-private case, by the margin and the modified combined margin and density assumptions, yielding the third error term of the theorem.

Compared to the previous result, the rate of convergence exhibits two key differences. The estimation error grows with $h^{-2d_a}$, instead of $h^{-d_a}$ which was the case for (13). Besides, the rate of convergence depends on $\gamma_2$, which is the modified combined margin and density condition parameter.

If for all $n \in \mathbb{N}^+$:

$$h_n \leq n^{\frac{\min(1,\gamma,\gamma_1) - \min(\gamma,\gamma_2)}{d_a(2\min(\gamma,\gamma_2) + 1 - \min(1,\gamma,\gamma_1))}},$$

then the second and fourth terms are dominated by the third term, hence for

$$h_n = (\sigma_Z^2/n)^{\frac{1}{2+2d_a}},$$

one has

$$\mathbb{E}\{L(\widetilde{D}_n)\} - L^* = O\left((\sigma_Z^2/n)^{\frac{1+\min\{\gamma,\gamma_2\}}{2+2d_a}}\right).$$

This means that for $\gamma \leq \gamma_2$ and $d = d_a$ we get the same *minimax optimal* rate as Berrett & Butucea (2019).

In the privatised case of non-binary classification, we use the privatised dataset $\{Z_{i,j,k}\}$, where

$$Z_{i,j,k} = \mathbb{I}_{\{Y_i=k\}}\mathbb{I}_{\{X_i \in A_{h,j}\}} + \sigma_Z\epsilon_{i,j,k}, \quad j = 1, \dots$$

Set

$$\widetilde{\nu}_{n,k}(A_{h,j}) = \frac{1}{n}\sum_{i=1}^{n} Z_{i,j,k},$$

and

$$\widetilde{D}_n(x) = \arg\max_{k} \widetilde{\nu}_{n,k}(A_{h,j}) \quad \text{for } x \in A_{h,j}.$$

Let us introduce the multi-class modified combined margin and density condition for privatisation. We say that $P_1, \dots, P_M$ satisfy the modified combined margin and density condition if there exists $h_2^*$ such that for all $h \in (0, h_2^*)$ we have for all $t > 0$:

$$\widetilde{G}_h(t) \doteq \int \mathbb{I}_{\{0 < f_h(x)(P_{(1)}(x) - P_{(2)}(x)) \leq t\}} \frac{1}{f_h(x)}\mu_a(dx) \leq c_2\, t^{\gamma_2} \tag{20}$$

with $c_2 > 0$ and $\gamma_2 \geq 0$.

In our final theorem, which is proved in Appendix A.7, we present the multi-class version of Theorem 3.3.

**Theorem 3.4.** *Assume that $X$ is bounded, $P_1, \dots, P_M$ satisfy the Lipschitz condition, Equation 3, the multi-class margin condition, Equation 16 with $\gamma \geq 0$, the multi-class combined margin and density condition, Equation 17 with $\gamma_1 \geq 0$ and the modified combined margin condition, Equation 20 with $\gamma_2 \geq 0$. Let $(h_n)$ be a monotonically decreasing sequence with zero limit. Then, we have*

$$\mathbb{E}\{L(\widetilde{D}_n)\} - L^* = O\big(M^2 h_n^{1+\gamma}\big) + O\left(M^2\left(\frac{1}{nh_n^{d_a}}\right)^{(\min\{1,\gamma,\gamma_1\}+1)/2}\right)$$
$$+ O\left(M^2\left(\frac{\sigma_Z^2}{nh_n^{2d_a}}\right)^{(1+\min\{\gamma,\gamma_2\})/2}\right) + O\left(\frac{M^2\sigma_Z}{nh_n^{d_a}}\right).$$

The effect of privatisation for multi-class classification is similar to the binary case. The right hand side grows quadratically with the number of classes. The variance of the privatisation only effects the third and fourth term. Typically the third term dominates the estimation error because of the extra 2 factor w.r.t. the bandwidth. Based on the theorem $h_n = (\sigma_Z^2/n)^{1/(2+d_a)}$ is an adequate choice to achieve similar rates as Berrett & Butucea (2019) if the margin parameter is dominant.

## 4 Discussion

In this paper we investigated both the binary and the multi-class versions of partitioning classification. Studying these methods can help better understanding a wide range of local averaging estimators, and it is directly relevant for various statistical and machine learning methods, including federated learning and ensemble-based approaches. One of our main contributions was that we weakened the strong density assumption, used in previous works, and proved novel convergence rates under the margin condition and a newly introduced combined margin and density condition. It was shown that the minimax optimal convergence rate, previously proved using SDA, can be achieved under much milder assumptions, refuting the conjecture that without SDA the convergence rate of the classification error probability can be arbitrarily slow.

We extended our results to the (locally differentially) private partitioning algorithm, which has no direct access to the data, only to its Laplace noise-perturbed version. We proved novel convergence rates under the margin condition, combined margin and density condition and the modified combined margin and density condition. As expected, the convergence rates of the nonprivate algorithms are faster than the corresponding privatised ones. Our theorems quantify the effect of privatisation by incorporating an extra rate of 2 for the bandwidth of the partitions and calibrating the dependence of the rate to the density of the inputs.

## Acknowledgements

The research of László Györfi was supported by the National Research, Development and Innovation Office (NKFIH) of Hungary under the 2023-1.1.1-PIACI-FÓKUSZ-2024-00051 funding scheme. The work of Balázs Cs. Csáji and Ambrus Tamás was also supported, in part, by the NKFIH, ADVANCED project no. 153 390, and by the European Commission through the DiGreeS project under grant no. 101178079.

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

## A Proofs

### A.1 Proof of Lemma 2.1

*Proof.* One has that

$$
\begin{aligned}
G_h(t) &= \int_{S_a} \mathbb{I}_{\{0<\sqrt{f_h(x)}|m(x)|\leq t\}} \frac{1}{\sqrt{f_h(x)}} \mu_a(dx) \\
&= \int_{B_\epsilon^*} \mathbb{I}_{\{0<\sqrt{f_h(x)}|m(x)|\leq t\}} \frac{1}{\sqrt{f_h(x)}} \mu_a(dx) + \int_{S_a \setminus B_\epsilon^*} \mathbb{I}_{\{0<\sqrt{f_h(x)}|m(x)|\leq t\}} \frac{1}{\sqrt{f_h(x)}} \mu_a(dx) \\
&\leq \int_{B_\epsilon^*} \mathbb{I}_{\{0<\sqrt{f_{\epsilon,min}}|m(x)|\leq t\}} \frac{1}{\sqrt{f_{\epsilon,min}}} \mu_a(dx) + \int_{S_a \setminus B_\epsilon^*} \mathbb{I}_{\{0<\sqrt{f_h(x)}\epsilon\leq t\}} \frac{1}{\sqrt{f_h(x)}} \mu_a(dx).
\end{aligned}
$$

By the definition of the margin condition,

$$
\begin{aligned}
\int_{B_\epsilon^*} \mathbb{I}_{\{0<\sqrt{f_{\epsilon,min}}|m(x)|\leq t\}} \frac{1}{\sqrt{f_{\epsilon,min}}} \mu_a(dx) &\leq \int \mathbb{I}_{\{0<|m(x)|\leq t/\sqrt{f_{\epsilon,min}}\}} \mu(dx)/\sqrt{f_{\epsilon,min}} \\
&= G^*\left(t/\sqrt{f_{\epsilon,min}}\right)/\sqrt{f_{\epsilon,min}} \leq c^* \left(t/\sqrt{f_{\epsilon,min}}\right)^\gamma/\sqrt{f_{\epsilon,min}}.
\end{aligned}
$$

For the notation

$$
H(s) := \mu_a(\{x : 0 < f_h(x) \leq s\}),
$$

one gets that

$$
\begin{aligned}
H(s) &= \sum_j \mu_a(\{x : 0 < f_h(x) \leq s, x \in A_{h,j}\}) \\
&= \sum_j \mu_a(\{x : 0 < \mu_a(A_{h,j})/h^{d_a} \leq s, x \in A_{h,j}\}) \\
&= \sum_j \mathbb{I}_{\{0<\mu_a(A_{h,j})/h^{d_a}\leq s\}} \mu_a(A_{h,j}) \\
&\leq \sum_j \mathbb{I}_{\{0<\mu_a(A_{h,j})\}} h^{d_a} \cdot s \\
&\leq const \cdot s.
\end{aligned}
\tag{21}
$$

Therefore,

$$
\begin{aligned}
\int_{S_a \setminus B_\epsilon^*} \mathbb{I}_{\{0<\sqrt{f_h(x)}\epsilon\leq t\}} \frac{1}{\sqrt{f_h(x)}} \mu_a(dx) &\leq \int_{S_a} \mathbb{I}_{\{0<\sqrt{f_h(x)}\leq t/\epsilon\}} \frac{1}{\sqrt{f_h(x)}} \mu_a(dx) \\
&= \int_0^{(t/\epsilon)^2} \frac{1}{\sqrt{s}} H(ds) \leq const \cdot t/\epsilon,
\end{aligned}
$$

and the lemma is proved. □

### A.2 Proof of Theorem 2.2

*Proof.* It is known that

$$L(D) - L^* = \int \mathbb{I}_{\{D(x) \neq D^*(x)\}} |m(x)| \mu(dx), \tag{22}$$

cf. Theorem 2.2 in (Devroye et al., 1996). For notational simplicity let $h = h_n$. For $x \in A_{h,j}$, we set

$$m_n(x) = \frac{\nu_n(A_{h,j})}{\mu(A_{h,j})}.$$

Because of (22), we have that

$$
\begin{aligned}
L(D_n) - L^* &= \int \mathbb{I}_{\{\text{sign } (m_n(x)) \neq \text{sign } (m(x))\}} |m(x)| \mu(dx) \\
&\leq \int \mathbb{I}_{\{|m_n(x) - m(x)| \geq |m(x)|\}} |m(x)| \mu(dx) \\
&\leq I_n + J_n,
\end{aligned}
$$

where

$$I_n = \int \mathbb{I}_{\{|\mathbb{E}\{m_n(x)\} - m(x)| \geq |m(x)|/2\}} |m(x)| \mu(dx)$$

and

$$J_n = \int \mathbb{I}_{\{|m_n(x) - \mathbb{E}\{m_n(x)\}| \geq |m(x)|/2\}} |m(x)| \mu(dx).$$

As the approximation error $I_n$,

$$I_n = \sum_j \int_{A_{h,j}} \mathbb{I}_{\{|\nu(A_{h,j}) - \mu(A_{h,j})m(x)| \geq \mu(A_{h,j})|m(x)|/2\}} |m(x)| \mu(dx),$$

where $\nu(A_{h,j}) = \mathbb{E}[Y \mathbb{I}_{\{X \in A_{h,j}\}}]$. The Lipschitz condition implies that

$$
\begin{aligned}
|\nu(A_{h,j}) - \mu(A_{h,j})m(x)| &\leq \left| \int_{A_{h,j}} m(z)\mu(dz) - \mu(A_{h,j})m(x) \right| \\
&\leq \int_{A_{h,j}} |m(z) - m(x)| \, \mu(dz) \\
&\leq C\sqrt{d}h\mu(A_{h,j}).
\end{aligned}
$$

This together with the margin condition yields that

$$I_n \leq \sum_j \int_{A_{h,j}} \mathbb{I}_{\{C\sqrt{d}h \geq |m(x)|/2\}} |m(x)| \mu(dx) \leq c^*(2C\sqrt{d}h)^{1+\gamma}, \tag{23}$$

and thus the bound on the approximation error in (13).

Concerning the estimation error $J_n$, we have that

$$
\begin{aligned}
\mathbb{E}\{J_n\} &= \sum_{A \in \mathcal{P}_h} \int_A \mathbb{P}\{|m_n(x) - \mathbb{E}\{m_n(x)\}| \geq |m(x)|/2\} |m(x)| \mu(dx) \\
&= \sum_{A \in \mathcal{P}_h} \int_A \mathbb{P}\{|\nu_n(A) - \nu(A)| \geq \mu(A)|m(x)|/2\} |m(x)| \mu(dx).
\end{aligned}
$$

Because of the CLT we have that

$$\sum_{A \in \mathcal{P}_h} \int_A \mathbb{P}\Big(|\nu_n(A) - \nu(A)| \geq \mu(A)|m(x)|/2\Big)|m(x)|\mu(dx)$$

$$\approx 2 \sum_{A \in \mathcal{P}_h} \int_A \Phi\left(-\sqrt{n}\frac{\mu(A)|m(x)|/2}{\sqrt{\mathbb{V}ar(Y\mathbb{I}_{\{X \in A\}})}}\right)|m(x)|\mu(dx), \tag{24}$$

(At the end of this subsection we show that the error term for this CLT approximation is of order $O(1/(nh_n^{d_a}))$.) Because of

$$\mathbb{V}ar(Y\mathbb{I}_{\{X \in A\}}) \leq \mathbb{E}[\mathbb{I}_{\{X \in A\}}] = \mu(A)$$

and up to the error term just mentioned, this implies

$$\mathbb{E}\{J_n\} \leq 2 \sum_{A \in \mathcal{P}_h} \int_A \Phi\left(-\sqrt{n}\sqrt{\mu(A)}|m(x)|/2\right)|m(x)|\mu(dx). \tag{25}$$

Next, we use the inequality

$$\Phi(-t) \leq e^{-t^2/2}\frac{1}{\sqrt{2\pi}}\frac{1}{t},$$

($t > 0$, cf. p. 179 in (Feller, 1957)). This together with (25) implies

$$\mathbb{E}\{J_n\} \leq 2 \sum_{A \in \mathcal{P}_h} \int_A \exp\left(-\frac{n}{8}\mu(A)m(x)^2\right) \frac{|m(x)|}{\sqrt{n}|m(x)|\sqrt{\mu(A)}/2}\mu(dx)$$

$$\leq \frac{4}{\sqrt{n}} \sum_{A \in \mathcal{P}_h} \int_A \exp\left(-\frac{n}{8}\mu(A)m(x)^2\right) \frac{1}{\sqrt{\mu(A)}}\mu(dx).$$

From decomposition (8) one gets that

$$\mathbb{E}\{J_n\} \leq \frac{4}{\sqrt{n}} \sum_{A \in \mathcal{P}_h} \mathbb{I}_{\{\mu(A) > h^{d_a}\}} \int_A \exp\left(-\frac{n}{8}m(x)^2\mu(A)\right) \mu(dx)\frac{1}{\sqrt{\mu(A)}}$$

$$+ \frac{4}{\sqrt{n}} \sum_{A \in \mathcal{P}_h} \mathbb{I}_{\{h^{d_a} \geq \mu(A) > 0\}} \int_A \exp\left(-\frac{n}{8}m(x)^2\mu(A)\right) \mu_a(dx)\frac{1}{\sqrt{\mu(A)}}$$

$$+ \frac{4}{\sqrt{n}} \sum_{A \in \mathcal{P}_h} \mathbb{I}_{\{h^{d_a} \geq \mu_s(A) > 0\}} \sqrt{\mu_s(A)}.$$

The discrete part can be handled as follows. Set $S_s$ is finite, therefore

$$\sum_{A \in \mathcal{P}_h} \mathbb{I}_{\{h^{d_a} \geq \mu_s(A) > 0\}} \leq \sum_{x \in S_s} \mathbb{I}_{\{h^{d_a} \geq \mu_s(\{x\}) > 0\}} = 0 \tag{26}$$

for $h$ small enough. (We note that the CLT approximation is not needed for the discrete part.)

Additionally, one has that

$$\frac{4}{\sqrt{n}} \sum_{A \in \mathcal{P}_h} \mathbb{I}_{\{\mu(A) > h^{d_a}\}} \int_A \exp\left(-\frac{n}{8}m(x)^2\mu(A)\right) \mu(dx)\frac{1}{\sqrt{\mu(A)}}$$

$$\leq \frac{4}{\sqrt{n}} \sum_{A \in \mathcal{P}_h} \mathbb{I}_{\{\mu(A) > h^{d_a}\}} \int_A \exp\left(-\frac{nh^{d_a}}{8}m(x)^2\right) \mu(dx)\frac{1}{\sqrt{h^{d_a}}}$$

$$\leq \frac{4}{\sqrt{nh^{d_a}}} \int \exp\left(-\frac{nh^{d_a}}{8}m(x)^2\right) \mu(dx).$$

For $a = nh^{d_a}/8$, partial integration together with the margin condition implies

$$\frac{4}{\sqrt{nh^{d_a}}} \int \exp\left(-nh^{d_a}m(x)^2/8\right)\mu(dx) = \frac{4}{\sqrt{nh^{d_a}}} \int_0^\infty e^{-as^2} G^*(ds)$$

$$= \frac{4}{\sqrt{nh^{d_a}}} 2a \int_0^\infty se^{-as^2} G^*(s)ds \leq \frac{4}{\sqrt{nh^{d_a}}} 2c^* a \int_0^\infty e^{-as^2} s^{1+\gamma}ds$$

$$= \frac{4}{\sqrt{nh^{d_a}}} 2c^* a^{-\gamma/2} \int_0^\infty e^{-u^2} u^{1+\gamma}du = O\left(\frac{1}{(nh^{d_a})^{(\gamma+1)/2}}\right), \tag{27}$$

where recall that $G^*(s) = \int \mathbb{I}_{\{0 < |m(x)| \leq t\}}\mu(dx)$. (Note that the CLT approximation with an error term $O(1/(nh_n^{d_a}))$ is justified by (31) below.)

For $a = nh^{d_a}/8$, the combined margin and density assumption implies that for all $0 < h \leq h_1^*$ we have

$$\frac{4}{\sqrt{n}} \sum_{A \in \mathcal{P}_h} \mathbb{I}_{\{h^{d_a} \geq \mu(A) > 0\}} \int_A \exp\left(-\frac{n}{8}m(x)^2\mu(A)\right)\mu_a(dx)\frac{1}{\sqrt{\mu(A)}}$$

$$\leq \frac{4}{\sqrt{n}} \sum_{A \in \mathcal{P}_h} \int_A \exp\left(-\frac{n}{8}m(x)^2\mu_a(A)\right)\mu_a(dx)\frac{1}{\sqrt{\mu_a(A)}}$$

$$= \frac{4}{\sqrt{n}} \int_{S_a} \exp\left(-nh^{d_a}m(x)^2 f_h(x)/8\right)\frac{1}{\sqrt{h^{d_a}f_h(x)}}\mu_a(dx)$$

$$= \frac{4}{\sqrt{nh^{d_a}}} \int_0^\infty e^{-as^2} G_h(ds) = \frac{4}{\sqrt{nh^{d_a}}} 2a \int_0^\infty se^{-as^2} G_h(s)ds$$

$$\leq \frac{4}{\sqrt{nh^{d_a}}} 2c_1 a \int_0^\infty e^{-as^2} s^{1+\gamma_1}ds = \frac{4}{\sqrt{nh^{d_a}}} 2c_1 a^{-\gamma_1/2} \int_0^\infty e^{-u^2} u^{1+\gamma_1}du$$

$$= O\left(\frac{1}{(nh_n^{d_a})^{(\gamma_1+1)/2}}\right), \tag{28}$$

(We note that the CLT approximation with an error term $O(1/(nh_n^{d_a}))$ is justified by (30) below.) (26), (27) and (28) together with (30) and (31) below yield the bound on the estimation error in (13). $\qquad\square$

*The CLT approximation error.*
For the CLT approximation in Equation 24, we need an upper bound. Put $Z_{i,A} = Y_i\mathbb{I}_{\{X_i \in A\}}$. We use the Berry-Esséen theorem and the normal approximation to upper bound this probability. Recall that because of the nonuniform Berry-Esséen theorem there exists a universal constant $0.4097 < c < 0.4785$ such that for all $a \in \mathbb{R}$ we have

$$\left|\mathbb{P}\left(\frac{\sqrt{n}}{\sigma_A n} \sum_{i=1}^n Z_{i,A} - \frac{\sqrt{n}}{\sigma_A}\mathbb{E}Z_{i,A} < a\right) - \Phi(a)\right| \leq \frac{c\varrho_A}{(1+|a|^3)\sigma_A^3\sqrt{n}},$$

where $\varrho_A = \mathbb{E}[|Z_{i,A} - \mathbb{E}Z_{i,A}|^3]$ and $\sigma_A^2 = \mathbb{V}ar(Z_{i,A})$, see (Esséen, 1956) and (Tyurin, 2010). Thus,

$$\mathbb{P}\left(\frac{\sqrt{n}}{\sigma_A n} \sum_{i=1}^n Z_{i,A} - \frac{\sqrt{n}}{\sigma_A}\mathbb{E}Z_{i,A} < a\right) \leq \Phi(a) + \frac{c\varrho_A}{(1+|a|^3)\sigma_A^3\sqrt{n}}.$$

It implies that

$$\mathbb{P}\left(\nu_n(A) - \nu(A) \geq \mu(A)|m(x)|/2\right)$$

$$= \mathbb{P}\left(\frac{\sqrt{n}}{\sigma_A}\nu_n(A) - \frac{\sqrt{n}}{\sigma_A}\nu(A) \geq \frac{\sqrt{n}}{\sigma_A}\mu(A)|m(x)|/2\right)$$

$$\leq \Phi\left(-\sqrt{n}\frac{\mu(A)|m(x)|/2}{\sigma_A}\right) + \frac{c\varrho_A}{(1+|\sqrt{n}\frac{\mu(A)|m(x)|/2}{\sigma_A}|^3)\sigma_A^3\sqrt{n}}.$$

Therefore, the error term in the approximation Equation 24 is equal to

$$\Delta := 2 \sum_{A \in \mathcal{P}_h} \int_A \frac{c\varrho_A}{(1 + |\sqrt{n}\frac{\mu(A)|m(x)|/2}{\sigma_A}|^3)\sigma_A^3\sqrt{n}} |m(x)|\mu(dx)$$

$$\leq 4 \sum_{A \in \mathcal{P}_h} \int_A \frac{c\varrho_A}{\sigma_A^2 n} \frac{\sqrt{n}\mu(A)|m(x)|/(2\sigma_A)}{(1 + |\sqrt{n}\mu(A)|m(x)|/(2\sigma_A)|^3)}\mu(dx)\frac{1}{\mu(A)}$$

$$\leq 4 \max_z \frac{z}{1 + z^3} \sum_{A \in \mathcal{P}_h} \frac{c\varrho_A}{\sigma_A^2 n}. \tag{29}$$

A simple consideration yields

$$\varrho_A = 8\mathbb{E}\left[\left(\frac{|Z_{i,A} - \mathbb{E}Z_{i,A}|}{2}\right)^3\right] \leq 8\mathbb{E}\left[\left(\frac{|Z_{i,A} - \mathbb{E}Z_{i,A}|}{2}\right)^2\right] = 2\sigma_A^2.$$

Therefore, by the boundedness of $X$,

$$\Delta \leq O\left(\frac{1}{n}\right) \sum_{A \in \mathcal{P}_h} 1 = O\left(\frac{1}{nh_n^d}\right).$$

If $\Delta$ is replaced by $\Delta'$ via replacing $\mu(dx)$ by $\mu_a(dx)$, then

$$\Delta' = O\left(\frac{1}{n}\right) \sum_{A \in \mathcal{P}_h} \frac{1}{\mu(A)}\mu_a(A) \leq O\left(\frac{1}{n}\right) \sum_{A \in \mathcal{P}_h} \mathbb{I}(\mu_a(A) > 0) = O\left(\frac{1}{nh_n^{d_a}}\right). \tag{30}$$

If $\Delta$ is modified by $\Delta''$ via inserting the factor $\mathbb{I}_{\{\mu(A) > h^{d_a}\}} \leq \mu(A)/h^{d_a}$ into the summands, then

$$\Delta'' = O\left(\frac{1}{n}\right) \sum_{A \in \mathcal{P}_h} \frac{1}{h^{d_a}}\mu(A) = O\left(\frac{1}{nh_n^{d_a}}\right). \tag{31}$$

### A.3  Proof of Theorem 2.3

Let

$$P_{n,k}(x) = \frac{\nu_{n,k}(A_{h,j})}{\mu(A_{h,j})}, \quad \text{if } x \in A_{h,j}.$$

The main ingredient of the proofs is the slight extension of (Györfi & Weiss, 2021, Lemma 8):

**Lemma A.1.** *Let $g_n$ be a plug-in rule with any estimates $P_{n,j}$ of $P_j$. For the notation*

$$\Delta_l(x) = P_{(1)}(x) - P_{(l)}(x),$$

*we have*

$$\mathbb{E}\{L(g_n)\} - L^* \leq \sum_{j=1}^M \sum_{l=2}^M J_{n,j,l}$$

*where*

$$J_{n,j,l} = \int \Delta_l(x)\mathbb{P}\{|P_{n,j}(x) - P_j(x)| \geq \Delta_l(x)/2\}\mu(dx). \tag{32}$$

*Proof.* As in the proof of Lemma 8 in (Györfi & Weiss, 2021),

$$\mathbb{E}\{L(g_n)\} - L^* = \int \mathbb{E}\{(P_{g^*(x)}(x) - P_{g_n(x)}(x))\mathbb{I}_{\{g^*(x)\neq g_n(x)\}}\}\mu(dx)$$

$$= \int \mathbb{E}\{(P_{g^*(x)}(x) - P_{g_n(x)}(x))\mathbb{I}_{\{P_{g^*(x)}(x)>P_{g_n(x)}(x)\}}\mathbb{I}_{\{P_{n,g_n(x)}(x)\geq P_{n,g^*(x)}(x)\}}\}\mu(dx).$$

If $g^*(x) = j$ and $g_n(x) = l$, then

$$\{P_{n,g_n(x)}(x) - P_{n,g^*(x)}(x) \geq 0\} = \{P_{n,l}(x) - P_l(x) + P_l(x) - P_{g^*(x)}(x) + P_j(x) - P_{n,j}(x) \geq 0\}$$

$$\subset \{|P_{n,j}(x) - P_j(x)| \geq (P_{g^*(x)}(x) - P_{g_n(x)}(x))/2\} \cup \{|P_{n,l}(x) - P_l(x)| \geq (P_{g^*(x)}(x) - P_{g_n(x)}(x))/2\}.$$

Therefore,

$$\mathbb{E}\{L(g_n)\} - L^*$$

$$\leq \sum_{j=1}^{M} \int \mathbb{E}\{(P_{g^*(x)}(x) - P_{g_n(x)}(x))\mathbb{I}_{\{|P_{n,j}(x)-P_j(x)|\geq(P_{g^*(x)}(x)-P_{g_n(x)}(x))/2\}}\}\mu(dx)$$

$$\leq \sum_{j=1}^{M}\sum_{l=2}^{M} \int (P_{(1)}(x) - P_{(l)}(x))\mathbb{E}\{\mathbb{I}_{\{|P_{n,j}(x)-P_j(x)|\geq(P_{(1)}(x)-P_{(l)}(x))/2\}}\}\mu(dx)$$

$$= \sum_{j=1}^{M}\sum_{l=2}^{M} \int \Delta_l(x)\mathbb{P}\{|P_{n,j}(x) - P_j(x)| \geq \Delta_l(x)/2\}\mu(dx).$$

$\square$

*Proof.* As above let us use the simplified notation $h = h_n$. We bound Equation 32 by

$$J_{n,k,l} \leq J_{n,k,l}^{(1)} + J_{n,k,l}^{(2)},$$

where

$$J_{n,k,l}^{(1)} = \int \Delta_l(x)\mathbb{P}\{|P_{n,k}(x) - \mathbb{E}\{P_{n,k}(x)\}| \geq \Delta_l(x)/4\}\mu(dx),$$

and

$$J_{n,k,l}^{(2)} = \int \Delta_l(x)\mathbb{I}_{\{|\mathbb{E}\{P_{n,k}(x)\}-P_k(x)|\geq\Delta_l(x)/4\}}\mu(dx).$$

Concerning the estimation error $J_{n,k,l}^{(1)}$, as in the proof of Theorem 2.2 we apply the CLT with the Berry-Esséen bound and then decomposition (8) and also (26). For $x \in A_{h,j}$, we have that

$$\mathbb{P}\{|P_{n,k}(x) - \mathbb{E}\{P_{n,k}(x)\}| \geq \Delta_l(x)/4\}$$

$$= \mathbb{P}\{|\nu_{n,k}(A_{h,j}) - \mathbb{E}\{\nu_{n,k}(A_{h,j})\}| \geq \mu(A_{h,j})\Delta_l(x)/4\}$$

$$\approx 2\Phi\left(-\sqrt{n}\frac{\mu(A_{h,j})|\Delta_l(x)|/4}{\sqrt{\mathbb{V}ar(Y\mathbb{I}_{\{X\in A_{h,j}\}})}}\right),$$

and henceforth, up to a term $O(1/(nh_n^{d_a}))$,

$$\sum_{A\in\mathcal{P}_h}\int_A \mathbb{P}\left(|\nu_{n,k}(A) - \mathbb{E}[\nu_{n,k}(A)]| \geq \mu(A)|\Delta_l(x)|/4\right)|\Delta_l(x)|\mu(dx)$$

$$\approx 2\sum_{A\in\mathcal{P}_h}\int_A \Phi\left(-\sqrt{n}\frac{\mu(A)|\Delta_l(x)|/4}{\sqrt{\mathbb{V}ar(Y\mathbb{I}_{\{X\in A\}})}}\right)|\Delta_l(x)|\mu(dx)$$

$$\leq \frac{8}{\sqrt{n}} \sum_{A \in \mathcal{P}_h} \int_A \exp\left(-\frac{n}{32}\mu(A)\Delta_l(x)^2\right) \frac{1}{\sqrt{\mu(A)}} \mu(dx)$$

$$\leq \frac{8}{\sqrt{n}} \sum_{A \in \mathcal{P}_h} \mathbb{I}_{\{\mu(A) > h^{d_a}\}} \int_A \exp\left(-\frac{n}{32}\Delta_l(x)^2\mu(A)\right) \mu(dx) \frac{1}{\sqrt{\mu(A)}}$$

$$+ \frac{8}{\sqrt{n}} \sum_{A \in \mathcal{P}_h} \mathbb{I}_{\{h^{d_a} \geq \mu_a(A) > 0\}} \int_A \exp\left(-\frac{n}{32}\Delta_l(x)^2\mu_a(A)\right) \mu_a(dx) \frac{1}{\sqrt{\mu(A)}}$$

$$+ \frac{8}{\sqrt{n}} \sum_{A \in \mathcal{P}_h} \mathbb{I}_{\{h^{d_a} \geq \mu_s(A) > 0\}} \sqrt{\mu_s(A)},$$

which is smaller than

$$\frac{8}{\sqrt{nh^{d_a}}} \int \exp\left(-\frac{nh^{d_a}}{32}\Delta_l(x)^2\right) \mu(dx) + \frac{8}{\sqrt{nh^{d_a}}} \int_{S_a} \exp\left(-\frac{nh^{d_a}}{32}\Delta_l(x)^2 f_h(x)\right) \frac{1}{\sqrt{f_h(x)}} \mu_a(dx),$$

for $h$ small enough. If $l \neq g^*(x)$, then $\Delta_l(x) \geq P_{(1)}(x) - P_{(2)}(x) =: \Delta(x)$ otherwise $\Delta_l(x) = 0$, therefore up to a Berry-Esséen error term we have

$$J_{n,k,l}^{(1)} \leq \frac{8}{\sqrt{nh^{d_a}}} \int \exp\left(-\frac{nh^{d_a}}{32}\Delta(x)^2\right) \mu(dx)$$

$$+ \frac{8}{\sqrt{nh^{d_a}}} \int_{S_a} \exp\left(-\frac{nh^{d_a}}{32}\Delta(x)^2 f_h(x)\right) \frac{1}{\sqrt{f_h(x)}} \mu_a(dx).$$

and also

$$\sum_{k=1}^{M} \sum_{l=1}^{M} J_{n,k,l}^{(1)} \leq \frac{8M^2}{\sqrt{nh^{d_a}}} \int \exp\left(-\frac{nh^{d_a}}{32}\Delta(x)^2\right) \mu(dx)$$

$$+ \frac{8M^2}{\sqrt{nh^{d_a}}} \int_{S_a} \exp\left(-\frac{nh^{d_a}}{32}\Delta(x)^2 f_h(x)\right) \frac{1}{\sqrt{f_h(x)}} \mu_a(dx). \tag{33}$$

Similarly to (27), the margin condition implies

$$\frac{8M^2}{\sqrt{nh^{d_a}}} \int \exp\left(-\frac{nh^{d_a}}{32}(P_{(1)}(x) - P_{(2)}(x))^2\right) \mu(dx)$$

$$= \frac{8M^2}{\sqrt{nh^{d_a}}} \int_0^\infty e^{-as^2} G^*(ds) = O\left(M^2(nh_n^{d_a})^{-(\gamma+1)/2}\right), \tag{34}$$

where $a = (nh^{d_a})/32$. As (28), the combined margin and density condition yields that

$$\frac{8M^2}{\sqrt{nh^{d_a}}} \int_{S_a} \exp\left(-\frac{nh^{d_a}}{32}(P_{(1)}(x) - P_{(2)}(x))^2 f_h(x)\right) \frac{1}{\sqrt{f_h(x)}} \mu_a(dx)$$

$$= \frac{8M^2}{\sqrt{nh^{d_a}}} \int_0^\infty e^{-as^2} G_h(ds) = O\left(M^2(nh_n^{d_a})^{-(\gamma_1+1)/2}\right). \tag{35}$$

Thus,

$$\sum_{k=1}^{M} \sum_{l=1}^{M} J_{n,k,l}^{(1)} = O\left(M^2(nh_n^{d_a})^{-(\gamma_1+1)/2}\right) + O\left(M^2(nh_n^{d_a})^{-(\gamma+1)/2}\right) + O\left(\frac{M^2}{nh_n^{d_a}}\right). \tag{36}$$

Concerning the approximation error $J_{n,j,l}^{(2)}$ (compare the proof of (23)), we have

$$J_{n,k,l}^{(2)} = \sum_j \int_{A_{h,j}} \Delta_l(x) \mathbb{I}_{\{|\mathbb{E}\{P_{n,k}(x)\} - P_k(x)| \geq \Delta_l(x)/4\}} \mu(dx)$$

$$= \sum_j \int_{A_{h,j}} \Delta_l(x) \mathbb{I}_{\{|\nu_k(A_{h,j})-\mu(A_{h,j})P_k(x)| \ge \mu(A_{h,j})\Delta_l(x)/4\}} |m(x)| \mu(dx).$$

The Lipschitz condition implies that

$$|\nu_k(A_{h,j}) - \mu(A_{h,j})P_k(x)| \le \left| \int_{A_{h,j}} P_k(z)\mu(dz) - \mu(A_{h,j})P_k(x) \right|$$

$$\le \int_{A_{h,j}} |P_k(z) - P_k(x)| \, \mu(dz) \le C\sqrt{d}h\mu(A_{h,j}),$$

where $\nu_k(A_{h,j}) = \int_{A_{h,j}} P_k(z)\mu(dz)$. This together with the margin condition yields that

$$J_{n,k,l}^{(2)} \le \sum_j \int_{A_{h,j}} \mathbb{I}_{\{C\sqrt{d}h \ge \Delta_l(x)/4\}} \Delta_l(x)\mu(dx) \le c^*(4C\sqrt{d}h)^{1+\gamma}, \tag{37}$$

and so

$$\sum_{k=1}^{M} \sum_{l=1}^{M} J_{n,k,l}^{(2)} = O\left(M^2 h_n^{1+\gamma}\right). \tag{38}$$

$\square$

### A.4 Proof of Lemma 3.1

*Proof.* One has that

$$\widetilde{G}_h(t) = \int_{S_a} \mathbb{I}_{\{0 < f_h(x)|m(x)| \le t\}} \frac{1}{f_h(x)} \mu_a(dx)$$

$$\le \int_{S_a} \mathbb{I}_{\{0 < \sqrt{f_{\epsilon,min}}\sqrt{f_h(x)}|m(x)| \le t\}} \frac{1}{\sqrt{f_{\epsilon,min}}\sqrt{f_h(x)}} \mu_a(dx)$$

By the definition of the combined margin and density condition,

$$\int_{S_a} \mathbb{I}_{\{0 < \sqrt{f_{\epsilon,min}}\sqrt{f_h(x)}|m(x)| \le t\}} \frac{1}{\sqrt{f_{\epsilon,min}}\sqrt{f_h(x)}} \mu_a(dx)$$

$$\le \int \mathbb{I}_{\left\{0 < \sqrt{f_h(x)}|m(x)| \le t/\sqrt{f_{\epsilon,min}}\right\}} \frac{1}{\sqrt{f_h(x)}} \mu(dx)/\sqrt{f_{\epsilon,min}}$$

$$= G_h\left(t/\sqrt{f_{\epsilon,min}}\right)/\sqrt{f_{\epsilon,min}} \le c_1\left(t/\sqrt{f_{\epsilon,min}}\right)^{\gamma_1}/\sqrt{f_{\epsilon,min}}.$$

$\square$

### A.5 Proof of Lemma 3.2

*Proof.* One has that

$$G_h(t) = \int_{S_a} \mathbb{I}_{\{0 < \sqrt{f_h(x)}|m(x)| \le t\}} \frac{1}{\sqrt{f_h(x)}} \mu_a(dx)$$

$$= \int_{\{f_h(x) \le 1\}} \mathbb{I}_{\{0 < \sqrt{f_h(x)}|m(x)| \le t\}} \frac{1}{\sqrt{f_h(x)}} \mu_a(dx) + \int_{\{f_h(x) > 1\}} \mathbb{I}_{\{0 < \sqrt{f_h(x)}|m(x)| \le t\}} \frac{1}{\sqrt{f_h(x)}} \mu_a(dx)$$

$$\le \int_{\{f_h(x) \le 1\}} \mathbb{I}_{\{0 < f_h(x)|m(x)| \le t\}} \frac{1}{f_h(x)} \mu_a(dx) + \int_{\{f_h(x) > 1\}} \mathbb{I}_{\{0 < |m(x)| \le t\}} \mu_a(dx) \le c_2 t^{\gamma_2} + c^* t^\gamma.$$

$\square$

### A.6 Proof of Theorem 3.3

*Proof.* Let $h = h_n$ be as above. For $x \in A_{h,j}$, we set

$$\widehat{m}_n(x) = \frac{\widetilde{\nu}_n(A_{h,j})}{\mu(A_{h,j})},$$

and

$$\widehat{m}'_n(x) = \frac{\frac{\sigma_Z}{n} \sum_{i=1}^n \epsilon_{i,j}}{\mu(A_{h,j})}.$$

Then,

$$\widehat{m}_n = \widehat{m}'_n + m_n,$$

and

$$\widetilde{D}_n(x) = \text{sign}\,(\widehat{m}_n(x)).$$

Because of (22), we have that

$$L(\widetilde{D}_n) - L^* = \int \mathbb{I}_{\{\text{sign}\,(\widehat{m}_n(x)) \neq \text{sign}\,(m(x))\}} |m(x)| \mu(dx)$$

$$\leq \int \mathbb{I}_{\{|\widehat{m}_n(x) - m(x)| \geq |m(x)|\}} |m(x)| \mu(dx)$$

$$\leq I_n + J_n + K_n,$$

where

$$I_n = \int \mathbb{I}_{\{|\mathbb{E}\{m_n(x)\} - m(x)| \geq |m(x)|/3\}} |m(x)| \mu(dx),$$

and

$$J_n = \int \mathbb{I}_{\{|m_n(x) - \mathbb{E}\{m_n(x)\}| \geq |m(x)|/3\}} |m(x)| \mu(dx),$$

and

$$K_n = \int \mathbb{I}_{\{|\widehat{m}'_n(x)| \geq |m(x)|/3\}} |m(x)| \mu(dx).$$

As in the proof of Theorem 2.2 one gets that

$$I_n + \mathbb{E}\{J_n\} = O\left(h_n^{1+\gamma}\right) + O\left(\left(\frac{1}{nh_n^{d_a}}\right)^{(\min\{1,\gamma,\gamma_1\}+1)/2}\right).$$

For the term $K_n$,

$$\mathbb{E}\{K_n\} = \int \mathbb{P}\{|\widehat{m}'_n(x)| \geq |m(x)|/3\} |m(x)| \mu(dx)$$

$$= \sum_j \int_{A_{h,j}} \mathbb{P}\left\{\left|\frac{\sigma_Z}{n} \sum_{i=1}^n \epsilon_{i,j}\right| \geq \mu(A_{h,j}) |m(x)|/3\right\} |m(x)| \mu(dx).$$

As above, the Berry-Esséen theorem yields

$$\mathbb{P}\left\{\left|\frac{\sigma_Z}{n} \sum_{i=1}^n \epsilon_{i,j}\right| \geq \mu(A_{h,j}) |m(x)|/3\right\}$$

$$= 2 \cdot \mathbb{P} \left\{ \frac{1}{\sqrt{n}} \sum_{i=1}^{n} \epsilon_{i,j} \geq \frac{\sqrt{n}\mu(A_{h,j})|m(x)|}{3\sigma_Z} \right\}$$

$$\leq 2\Phi\left( -\frac{\sqrt{n}\mu(A_{h,j})|m(x)|}{3\sigma_Z} \right) + \frac{2c\varrho_\varepsilon}{\sigma_\varepsilon^3 \left( 1 + \left| \frac{\sqrt{n}\mu(A_{h,j})|m(x)|}{3\sigma_Z} \right|^3 \right) \sqrt{n}}. \tag{39}$$

Similarly to Equation 29 the error of CLT is essentially dominated by the first term in Equation 39, since

$$\Delta = 2\sum_j \int_{A_{h,j}} \frac{c\varrho_\varepsilon}{\sigma_\varepsilon^3 \left( 1 + \left| \frac{\sqrt{n}\mu(A_{h,j})|m(x)|}{3\sigma_Z} \right|^3 \right) \sqrt{n}} |m(x)|\mu(dx)$$

$$\leq 6\max_z \frac{z}{1+z^3} \sum_j \frac{c\varrho_\varepsilon \sigma_Z}{\sigma_\varepsilon^3 n} = \frac{\tilde{c}\sigma_Z}{nh^d}$$

by the boundedness of $X$. Because of Equation 30 and Equation 31 this can be strenghened to $O(\sigma_Z/(nh^{d_a}))$.

Using the decomposition of (8), as in the proof of Theorem 2.2, along with Equation 26, Equation 27 and Equation 28 one obtains

$$2\sum_j \int_{A_{h,j}} \Phi\left( -\frac{\sqrt{n}\mu(A_{h,j})|m(x)|}{3\sigma_Z} \right) |m(x)|\mu(dx)$$

$$\leq 2\sum_j \int_{A_{h,j}} \frac{3\sigma_Z}{\sqrt{n}\mu(A_{h,j})|m(x)|} \exp\left( -\frac{n\mu(A_{h,j})^2 m(x)^2}{18\sigma_Z^2} \right) |m(x)|\mu(dx).$$

$$\leq 2\sum_j \mathbb{I}_{\{h^{d_a} < \mu(A_{h,j})\}} \int_{A_{h,j}} \frac{3}{\sqrt{n\mu(A_{h,j})^2/\sigma_Z^2}} \exp\left( -\frac{n\mu(A_{h,j})^2 m(x)^2}{18\sigma_Z^2} \right) \mu(dx)$$

$$+ 2\sum_j \mathbb{I}_{\{h^{d_a} \geq \mu(A_{h,j}) > 0\}} \int_{A_{h,j}} \frac{3}{f_h(x)\sqrt{nh^{2d_a}/\sigma_Z^2}} \exp\left( -\frac{nf_h(x)^2 h^{2d_a} m(x)^2}{18\sigma_Z^2} \right) \mu_a(dx)$$

$$\leq 2\int \frac{3}{\sqrt{nh^{2d_a}/\sigma_Z^2}} \exp\left( -\frac{nh^{2d_a} m(x)^2}{18\sigma_Z^2} \right) \mu(dx)$$

$$+ 2\int \frac{3}{f_h(x)\sqrt{nh^{2d_a}/\sigma_Z^2}} \exp\left( -\frac{nf_h(x)^2 h^{2d_a} m(x)^2}{18\sigma_Z^2} \right) \mu_a(dx)$$

for $h$ small enough. By the margin condition

$$\int \frac{3}{\sqrt{nh^{2d_a}/\sigma_Z^2}} \exp\left( -\frac{nh^{2d_a} m(x)^2}{18\sigma_Z^2} \right) \mu(dx) = O\left( \left( \frac{\sigma_Z^2}{nh_n^{2d_a}} \right)^{(\gamma+1)/2} \right).$$

and by the modified combined margin and density condition

$$\int \frac{3}{f_h(x)\sqrt{nh^{2d_a}/\sigma_Z^2}} \exp\left( -\frac{nf_h(x)^2 h^{2d_a} m(x)^2}{18\sigma_Z^2} \right) \mu_a(dx) = O\left( \left( \frac{\sigma_Z^2}{nh_n^{2d_a}} \right)^{(\gamma_2+1)/2} \right).$$

In conclusion, we have

$$\mathbb{E}(L(\tilde{g}_n)) - L^* = O\left( h^{1+\gamma} \right) + O\left( \left( \frac{1}{nh_n^{d_a}} \right)^{(\min\{1,\gamma,\gamma_1\}+1)/2} \right)$$

$$+ O\left( \left( \frac{\sigma_Z^2}{nh^{2d_a}} \right)^{(\gamma+1)/2} \right) + O\left( \left( \frac{\sigma_Z^2}{nh^{2d_a}} \right)^{(\gamma_2+1)/2} \right) + O\left( \frac{\sigma_Z}{nh^{d_a}} \right).$$

$\square$

## A.7 Proof of Theorem 3.4

*Proof.* Again, we bound Equation 32 by

$$J_{n,k,l} \leq J_{n,k,l}^{(1)} + J_{n,k,l}^{(2)} + J_{n,k,l}^{(3)},$$

where

$$J_{n,k,l}^{(1)} = \int \Delta_l(x)\mathbb{P}\{|P_{n,k}(x) - \mathbb{E}\{P_{n,k}(x)\}| \geq \Delta_l(x)/6\}\mu(dx),$$

and

$$J_{n,k,l}^{(2)} = \int \Delta_l(x)\mathbb{I}_{\{|\mathbb{E}\{P_{n,k}(x)\} - P_k(x)| \geq \Delta_l(x)/6\}}\mu(dx),$$

and

$$J_{n,k,l}^{(3)} = \int \Delta_l(x)\mathbb{P}\{|Q_{n,k}(x)| \geq \Delta_l(x)/6\}\mu(dx),$$

with

$$Q_{n,k}(x) = \frac{\frac{\sigma_Z}{n}\sum_{i=1}^n \epsilon_{i,j,k}}{\mu(A_{h,j})}, \quad \text{if } x \in A_{h,j}.$$

As in (36) and (38), up to a Berry-Esséen bound

$$\sum_{k=1}^M \sum_{l=1}^M J_{n,k,l}^{(1)} = O\left(M^2(nh_n^{d_a})^{-(\gamma_1+1)/2}\right) + O\left(M^2(nh_n^{d_a})^{-(\gamma+1)/2}\right),$$

and

$$\sum_{k=1}^M \sum_{l=1}^M J_{n,k,l}^{(2)} = O\left(M^2 h_n^{1+\gamma}\right).$$

Again, the CLT yields that

$$\sum_j \int_{A_{h,j}} \Delta_l(x)\mathbb{P}\left\{\left|\frac{\sigma_Z}{n}\sum_{i=1}^n \epsilon_{i,j,k}\right| \geq \mu(A_{h,j})\Delta_l(x)/6\right\}\mu(dx)$$

$$\leq \sum_j \int_{A_{h,j}} \frac{6\Delta_l(x)}{\sqrt{n}\mu(A_{h,j})\Delta_l(x)} \exp\left(-\frac{n\mu(A_{h,j})^2\Delta_l(x)^2/6^2}{4\sigma_Z^2}\right)\mu(dx) + O\left(\frac{\sigma_Z}{nh_n^{d_a}}\right)$$

$$\leq \sum_j \int_{A_{h,j}} \frac{6}{\sqrt{n}\mu(A_{h,j})} \exp\left(-\frac{n\mu(A_{h,j})^2\Delta(x)^2}{144\sigma_Z^2}\right)\mu(dx) + O\left(\frac{\sigma_Z}{nh_n^{d_a}}\right).$$

For $h$ small enough, this together with the arguments in (26), (27), (28), Equation 33 and Equation 34 imply

$$J_{n,k,l}^{(3)} \leq 2\sum_j \mathbb{I}_{\{h^{d_a} < \mu(A_{h,j})\}} \int_{A_{h,j}} \frac{6}{\sqrt{n}\mu(A_{h,j})} \exp\left(-\frac{n\mu(A_{h,j})^2\Delta(x)^2}{144\sigma_Z^2}\right)\mu(dx)$$

$$+ 2\sum_j \mathbb{I}_{\{h^{d_a} \geq \mu(A_{h,j}) > 0\}} \int_{A_{h,j}} \frac{6}{\sqrt{n}\mu(A_{h,j})} \exp\left(-\frac{n\mu_a(A_{h,j})^2\Delta(x)^2}{144\sigma_Z^2}\right)\mu_a(dx)$$

up to an $O\left(\frac{\sigma_Z}{nh_n^{d_a}}\right)$ term, and henceforth

$$\sum_{k=1}^M \sum_{l=1}^M J_{n,k,l}^{(3)} = O\left(M^2\left(\frac{\sigma_Z^2}{nh_n^{2d_a}}\right)^{(1+\min(\gamma,\gamma_2))/2}\right) + O\left(\frac{M^2\sigma_Z}{nh_n^{d_a}}\right).$$

$\square$

