# OpenReview forum: "On Rate-Optimal Partitioning Classification from Observable and from Privatised Data"
_TMLR — Accepted by TMLR_

### Review · Reviewer_zar2 · 2026-03-20

**Summary Of Contributions:**

This paper proposes a novel assumption to study partitioning classification methods, and applies it to locally differentially private classification.
Specifically, the proposed assumption, called *combined margin and density condition*, relaxes the classical *strong density assumption*:
- the strong density assumption assumes that each cell of the partition contains a large enough fraction of the data distribution (w.r.t. Lebesgue measure);
- in contrast, the proposed assumption combines the data distribution density with the classifier's margin, allowing for smaller data density in regions where the model is very confident.
Intuitively, this assumption builds on the idea that regions of the space where the classifier is very confident are much less likely to influence the classifier's decision, and do not impact the consistency results as much.
As such, this new assumption mitigates important limits of the strong density assumption; in particular, it allows to handle continuous data distributions whose support is $\mathbb{R}^d$, which do not satisfy the strong density assumption.

Based on this assumption, a consistency theorem can be derived for a much broader class of problems, and with a rate matching the ones obtained under the strong density assumption.
This refutes the conjecture that the strong density assumption is required to obtain consistency guarantees for partitioning classification methods.

Application to partitioning classification with local differential privacy is then proposed.
In particular, the novel assumption allows for handling distributions with full support on $\mathrm{R}^d$, allowing for handling noise from LDP, which is typically not bounded.
This allows to derive consistency results for any level of privacy (and thus, noise variance), which had only been derived for privacy budget going to infinity as the size of the partitions decreased before.

Furthermore, the proposed setting handles mixtures of continuous and discrete data distribution, and is also extended to the multi-class setting, both for the non-private and the private variants.

**Additional Comments:**

This goes beyond the scope of the paper, but could the observations derived in this manuscript lead to developing methods that are more consistent by adapting the way the partitions are made? Or by trying to maximize the margin in regions of high density? In general, would this assumption be applicable in methods where the partition is learned from a dataset rather than predefined?

This paper is purely theoretical, but a numerical study on cases where a strong density assumption does not hold but the convergence rate remains similar, would give a nice illustration of the theoretical results. Yet, I do not believe this should prevent from accepting the paper, which already has strong and original contributions.

**Audience:**

Yes

**Audience Explanation:**

This paper studies the theory of partitioning classification methods and introduces a novel assumption that is not trivial to define, while capturing a very intuitive idea that regions where data is scarce should not impact the predictions if the margin of the classifier is large at that point.

Remarkably, this novel assumption allows for deriving novel consistency results for locally differentially private partitioning classification method, removing a very strong limitation from previous works, which required the privacy budget to grow very fast to infinity, giving limited privacy guarantees.

**Broader Impact Concerns:**

Not applicable: this is a purely theoretical paper on the consistency of partitionning classification methods, with and without LDP guarantees.

**Claims And Evidence:**

Yes

**Claims Explanation:**

All results are given with the corresponding proof, and the rates obtained seem coherent with previous rates from prior work.

No numerical evidence is given, but this is not the goal of the paper.

**Requested Changes:**

The ideas developed in this paper are very interesting and seem to be correct, at least from my limited knowledge of this area.

There are a few points that could be improved regarding the presentation of the results:
- The idea behind the novel assumption that is proposed in this manuscript seems intuitive, but its transcription in mathematical terms is not straightforward: could the author comment more on the specific design choice in Equation (10)?
- Specifically, it seems that this design choice directly comes from the need to bound terms that appear in the error, which are handled more carefully than with the rougher strong density assumption. It seems that providing a proof sketch, showing the part of the derivations where this assumption is useful, and why it improves the result, would make it clearer why this is the right choice of assumption. This would give much clearer insights than the simplistic examples, which give some intuition but are described a bit too extensively.
- A different assumption is required for the LDP case: is it possible to derive a rate in the non-LDP case with this assumption? Why is this needed to prove the result in this case? Again, giving a proof sketch explaining where these terms appear would benefit the paper's readability.

These questions are not crucial for paper acceptance in my opinion, and the contributions presented are interesting per se, but this would make it easier to read the paper.

---

> ### Author Response · Authors · 2026-04-12
> **Response to Reviewer zar2 (1)**
>
> Thank you for your insightful review. Please find below our detailed answers.
>
> > The idea behind the novel assumption that is proposed in this manuscript seems intuitive, but its transcription in mathematical terms is not straightforward: could the author comment more on the specific design choice in Equation (10)? Specifically, it seems that this design choice directly comes from the need to bound terms that appear in the error, which are handled more carefully than with the rougher strong density assumption. It seems that providing a proof sketch, showing the part of the derivations where this assumption is useful, and why it improves the result, would make it clearer why this is the right choice of assumption. This would give much clearer insights than the simplistic examples, which give some intuition but are described a bit too extensively.
>
> Thanks for the specific suggestion. We will include a more detailed motivation for Equation (10) in the revised manuscript. Intuitively, Equation (10) let us control the errors in those cells that lie near the decision boundary or in low density regions. We will explicitly refer to Equation (27), where the full technical derivation is provided.
>
> > A different assumption is required for the LDP case: is it possible to derive a rate in the non-LDP case with this assumption?
>
> In Lemma 3.2, it is proved that the margin condition together with the modified margin and density condition implies the combined margin and density condition. This allows us to bound the non-LDP rate under the LDP assumptions. Moreover, the combined margin and density parameter can be upper-bounded by the minimum of the margin parameter and the modified margin and density parameter. On the other hand, the modified condition is more restrictive in this case, that is why we use slightly different conditions for the two cases.
>
> > Why is this needed to prove the result in this case? Again, giving a proof sketch explaining where these terms appear would benefit the paper's readability.
>
> The additional factors are required to control the privatization terms. We will explicitly refer to Equation (38), where the Berry–Esséen theorem is applied. The slower convergence rate primarily arises from the scaling: in the Berry–Esséen bound, the variance remains constant, so the bound depends on $\mu(A_{h,j})$ rather than on $\sqrt{\mu(A_{h,j})}$.
>
> > This goes beyond the scope of the paper, but could the observations derived in this manuscript lead to developing methods that are more consistent by adapting the way the partitions are made?
>
> This is an interesting research direction, which goes beyond the scope of the present paper. There are several works on the consistency of data-dependent partitioning (e.g., using Voronoi cells, random forests or histogram rules), but these do not address the optimal rate of convergence. See, for example:
>
> [1] G. Biau, L. Devroye, and G. Lugosi. Consistency of random forests and other averaging classifiers. Journal of Machine Learning Research, 9:2015–2033, 2008.
>
> [2] L. Breiman, J. Friedman, C. J. Stone and R.A. Olshen. Classification and Regression Trees. Taylor and Francis, 1984.
>
> [3] L. Devroye, L. Györfi and G. Lugosi, A Probabilistic Theory of Pattern Recognition, Springer–Verlag, New York, 1996.
>
> [4] G. Lugosi and A. Nobel. Consistency of data-driven histogram methods for density estimation and classification, Annals of Statistics 24:687–706, 1996.
>
> > Or by trying to maximize the margin in regions of high density?
>
> Thank you, it is a compelling idea. In general, controlling the convergence rate of data-dependent partitioning rules is challenging, as it requires accounting for the additional randomness compared to deterministic partitions. LDP bounds can be established for simpler cases, such as Voronoi cells. Estimating key parameters in this setting, such as the margin, the combined margin and density, and the modified combined margin and density, remains very difficult. These go beyond the scope of our current study, but it is indeed a promising future research direction.
>
> > In general, would this assumption be applicable in methods where the partition is learned from a dataset rather than predefined?
>
> Establishing convergence rates for data-driven partitioning methods is an active research direction. When partitions are learned from data, certain assumptions, such as those on margin or combined margin and density, may require refinement to properly account for the additional randomness introduced by the data-driven construction. We are planning to investigate how such modifications impact the theoretical guarantees.

---

> > ### Author Response · Authors · 2026-04-12
> > **Response to Reviewer zar2 (2)**
> >
> > > This paper is purely theoretical, but a numerical study on cases where a strong density assumption does not hold but the convergence rate remains similar, would give a nice illustration of the theoretical results. Yet, I do not believe this should prevent from accepting the paper, which already has strong and original contributions.
> >
> > Thank you for the supportive comment. We agree that a numerical study exploring cases where the strong density assumption (SDA) is relaxed would be an interesting illustration. However, our primary goal was to provide a rigorous theoretical analysis of relaxing the SDA, both in the classical and the privatized settings, hence we have focused exclusively on the mathematical derivations. The experimental study would be particularly valuable for more directly applicable methods, like federated learning, which we leave for future work.

---

### Review · Reviewer_uQ8A · 2026-04-07

**Summary Of Contributions:**

The paper considers the problem of (binary) classification in the d-dim euclidean space under the Strong Density Assumption [Audibert and Tsybakov]. SDA essentially states that in each A_{h,j}, the density must be lower-bounded by at least some constant - a very strong requirement as it e.g. essentially rules out „continous rensities on R^d“ which includes the classic discriminant analysis setting. The authors are able to relax the assumption and introduce a new assumption called _combined margin and densities_ and show that this is already sufficient ot obtain similar convergence rates as under SDA. Furthermore, they prove some relations between those assumptions.

Furthermore, the authors constructively disprove a conjecture due to [Berett and Butucea] that conjectured that SDA is essentially necessary to get fast convergence (in the LDP setting).

In general, the paper is well written from the short glances at the proofs a could take, they seems to be clean, very technical and correct. This is a pure theory paper, so experiments are not necessary.

**Audience:**

Yes

**Audience Explanation:**

The paper directly builds up on recent work on private classification [BB19] and in general on a whole line of work on classification. The results itself that shows that previous assumptions were essentially too strong are nice and interesting to the privacy, ML and statistics communities.

**Broader Impact Concerns:**

N/A. This is a pure theory paper.

**Claims And Evidence:**

Yes

**Claims Explanation:**

Every claim is justified with a proof in the appendix. The structure of the paper is very clear and thought out. I did not find time to check all of them, but from what I read, they seem to be correct.

**Requested Changes:**

I might have missed it, but can you clarify the constraint on the sensitivity in the LDP setting? I guess these kind of questions are discussed in [BB19], but it would be nice to have some discussion in the paper as well. Can you say something about different noise distributions (e.g. Gaussian?).

---

> ### Author Response · Authors · 2026-04-12
> **Response to Reviewer uQ8A**
>
> Thank you for your work and for finding our results relevant to the privacy, machine learning and statistics communities. Our detailed responses to your questions are provided below.
>
> > Can you clarify the constraint on the sensitivity in the LDP setting? I guess these kind of questions are discussed in [BB19], but it would be nice to have some discussion in the paper as well.
>
> Formally, a mechanism satisfies $\alpha$-LDP if, for any two possible inputs, the likelihood ratio of the corresponding privatized outputs is uniformly bounded. A brief introduction to $\alpha$-LDP is given in Section 3. The Laplace mechanism is particularly well suited in this context, as its exponential ($L_1$-based) form directly aligns with the likelihood ratio constraint, yielding exact privacy guarantees that are easy to calibrate. We will revise the manuscript to further clarify why the Laplace distribution is a natural choice in the LDP setting and refer to [1].
>
> [1] C. Dwork, F. McSherry, K. Nissim and A. Smith, Calibrating Noise to Sensitivity in Private Data Analysis. Journal of Privacy and Confidentiality, Vol. 7, No. 3, pp. 17-51, 2016
>
>  > Can you say something about different noise distributions (e.g. Gaussian)?
>
> LDP requires pointwise likelihood ratio control, which is harder to satisfy with Gaussian tails without careful calibration. Gaussian perturbation is more commonly used in the centralized differential privacy model, where it provides $(\varepsilon, \delta)$-privacy guarantees [2]. Consequently, Laplace-type or discrete mechanisms are typically preferred in LDP, due to their theoretical tractability and optimality properties in several statistical problems [3].
>
> [2] B. Balle and X. Y. Wang, Improving the Gaussian Mechanism for Differential Privacy: Analytical Calibration and Optimal Denoising. In International Conference on Machine Learning, pp. 394-403, 2018, PMLR
>
> [3] C. Dwork and A. Roth, The Algorithmic Foundations of Differential Privacy. Foundations and trends® in Theoretical Computer Science, 9(3-4), 2014

---

### Review · Reviewer_pZ3x · 2026-04-07

**Summary Of Contributions:**

This paper studies partitioning classification for high-dimensional data. The main contributions and strengths are the following,
1. The authors prove a new result on the convergence rate of the optimality gap of classification error ($E[L]-L^*$) without relying on the strong density assumption (SDA). The new result improves the best known rate without SDA for some examples while simultaneously recovering optimal error rates proved with the SDA assumption.
2. A nice aspect about the convergence rate proved is that it depends on the intrinsic dimension when the data is distributed over a low-dimensional space instead of the ambient dimension, which is promising from a practical perspective
3. The same machinery is also used to study the partitioning classification under local differential privacy.

Weaknesses
1. The main weakness is that the examples where the new bound improves upon prior SDA independent bounds are very specific. The paper could be strengthened by giving examples for a more general class of data distributions that one may encounter in practical scenarios. For example, the authors mentioned that Gaussian distribution is an important example that do not satisfy SDA, but it seems that the paper does not present improved convergence rates for Gaussian distribution.
2. Another minor weakness is a lack of simulation experiments. Even though it is often not necessary for theoretical papers, a simulation result could validate the theoretical claims and suggest a practical implications.

**Audience:**

Yes

**Audience Explanation:**

The paper should be interesting for researchers working on the theoretical aspects of machine learning as well as statistics.

**Broader Impact Concerns:**

The paper is theoretical and does not need a broader impact statement to address the potential ethical concerns.

**Claims And Evidence:**

Yes

**Claims Explanation:**

1. When arguing the limitations of the strong density assumption (SDA), the authors provide specific examples where SDA do not hold and existing approach yields sub-optimal convergence rates.
2. The authors compare the new bound with previous results using specific and well-explained examples.

**Requested Changes:**

These suggestions mainly serve to strengthen the paper.
1. Clearly state in the beginning of Section 2 that we care about the convergence rate of $E[L]-L^*$. Even though the definition is standard and used in other parts of the paper, for better presentation it would be nice to clearly state this upfront.
2. If possible, provide the convergence rate for Gaussian distributions as a result of the new proposed bound, or explain the technical difficulty in analyzing Gaussian distributions.
3. Add a brief discussion or remark about the computational aspects of the optimal estimate, namely computational cost and whether it relies on distributional assumptions.
4. If applicable, include suitable simulation experiments to verify the theoretical claims.

---

> ### Author Response · Authors · 2026-04-12
> **Response to Reviewer pZ3x**
>
> We are grateful for your constructive review. We provide our detailed responses to your questions below.
>
> > Clearly state in the beginning of Section 2 that we care about the convergence rate of $\mathbb{E}[L]- L^*$. Even though the definition is standard and used in other parts of the paper, for better presentation it would be nice to clearly state this upfront.
>
> Thank you for your suggestion. We will include such a description in the final manuscript to improve the presentation.
>
> > If possible, provide the convergence rate for Gaussian distributions as a result of the new proposed bound, or explain the technical difficulty in analyzing Gaussian distributions.
>
> Gaussian distributions are one example for which the SDA does not hold and we aimed to highlight this in the manuscript. Nevertheless, SDA can be restrictive even for bounded distributions. This will be emphasized in the final manuscript.
>
> In our analysis, we assume that the distribution of $X$ is bounded. It is required when bounding the approximation error via the central limit theorem. As a result, our theorem does not directly apply to Gaussian distributions. To avoid potential misunderstandings, we will add a clarification about boundedness following the first theorem in the final manuscript.
>
> > Add a brief discussion or remark about the computational aspects of the optimal estimate, namely computational cost and whether it relies on distributional assumptions.
>
> Thank you for the suggestion. In the revised manuscript, we will include a remark on the computational costs.
>
> For the nonprivate partitioning classifier, the primary computational burden arises from the number of cells that must be stored, which is directly influenced by the support of the input space. For this simple local averaging variant, the time complexity scales linearly with the number of observations.
>
> The privatized version is computationally more expensive. For privatization, the sample size is effectively multiplied by the number of cells. This increases both the storage requirements and the time complexity proportionally to the number of cells.
>
> On the other hand, we would like to emphasize that the primary objective of this work is not to propose a novel, computationally optimized algorithm, but rather to improve the theoretical understanding of local averaging mechanisms and to establish stronger foundations for their convergence properties.
>
> > If applicable, include suitable simulation experiments to verify the theoretical claims.
>
> Although numerical experiments could offer further verification, the present work is intentionally focused on the underlying technical developments. We believe this provides a solid foundation for future empirical investigations.

---

### Decision · Action_Editor_wPmw · 2026-05-31

**Recommendation:** Accept as is

**Additional Comments:**

The reviewers unanimously support acceptance, highlighting the novelty and relevance of the theoretical contributions, as well as the clarity of the presentation.

**Audience:**

Yes

**Audience Explanation:**

The paper addresses the problem on partitioning classification, which is somewhat nice but certainly relevant to TMLR's audience.

**Claims And Evidence:**

Yes

**Claims Explanation:**

This is a theory paper. All claims are supported by theorems along with their proof, and the presentation of the results are clear and accurate.